# Effectiveness of Percutaneous Nephrolithotomy, Retrograde Intrarenal Surgery, and Extracorporeal Shock Wave Lithotripsy for Treatment of Renal Stones: A Systematic Review and Meta-Analysis

**DOI:** 10.3390/medicina57010026

**Published:** 2020-12-30

**Authors:** Chan Hee Kim, Doo Yong Chung, Koon Ho Rha, Joo Yong Lee, Seon Heui Lee

**Affiliations:** 1Department of Nursing Science, College of Nursing, Gachon University, Incheon 22212, Korea; potbeau@gmail.com; 2Department of Urology, Inha University School of Medicine, Incheon 22212, Korea; wjdendyd@gmail.com; 3Department of Urology, Severance Hospital, Urological Science Institute, Yonsei University College of Medicine, Seoul 03722, Korea; khrha@yuhs.ac; 4Center of Evidence Based Medicine, Institute of Convergence Science, Yonsei University, Seoul 03722, Korea

**Keywords:** urolithiasis, percutaneous nephrolithotomy (PCNL), retrograde intrarenal surgery (RIRS), extracorporeal shock wave lithotripsy (SWL)

## Abstract

*Background and objectives*: To perform a updated systematic review and meta-analysis comparing effectiveness of percutaneous nephrolithotomy (PCNL), retrograde intrarenal surgery (RIRS), and extracorporeal shock wave lithotripsy (ESWL) for treatment of renal stones (RS). *Materials and Methods*: A total of 37 studies were included in this systematic review and meta-analysis about effectiveness to treat RS. Endpoints were stone-free rates (SFR), incidence of auxiliary procedure, retreatment, and complications. We also conducted a sub-analysis of ≥2 cm stones. *Results*: First, PCNL had the highest SFR than others regardless of stone sizes and RIRS showed a higher SFR than ESWL in <2 cm stones. Second, auxiliary procedures were higher in ESWL than others, and it did not differ between PCNL and RIRS. Finally, in <2 cm stones, the retreatment rate of ESWL was higher than others. RIRS required significantly more retreatment procedures than PCNL in ≥2 cm stones. Complication was higher in PCNL than others, but there was no statistically significant difference in complications between RIRS and PCNL in ≥2 cm stones. For ≥2 cm stones, PCNL had the highest SFR, and auxiliary procedures and retreatment rates were significantly lower than others. *Conclusions*: We suggest that PCNL is a safe and effective treatment, especially for large RS.

## 1. Introduction

Urolithiasis, a major clinical and economic burden for healthcare systems [1], causes severe pain in the flank or abdomen that may be accompanied by blood in the urine, vomiting, or painful urination. In addition, urinary stones have a 1-year recurrence rate of 7% and 10-year recurrence rate of 50% [2]. Although urinary stones have been successfully treated, these high recurrence rates make urolithiasis an important health issue requiring additional treatments [3]. 

As minimally invasive technologies develop, percutaneous nephrolithotomy (PCNL), retrograde intrarenal surgery (RIRS), and extracorporeal shock wave lithotripsy (ESWL) are the most commonly performed treatments for kidney stones [4,5]. The European Association of Urology (EAU) guidelines of urolithiasis suggests various treatments by stone type and size [6]. Unlike the EAU guidelines, ESWL is currently used to treat most cases since it can be performed easily without hospitalization [7]. However, ESWL often requires multiple treatments because of poor clearance and leads to surgical delays or an increasing burden of expenses [8]. 

Some systematic reviews have compared renal stone treatments. Our study expands on the existing systematic reviews by addressing the most comprehensive and updated data. Some studies [9,10,11] compared just two kinds of treatments [9,10,11], while others [4,12] compared all three but not by stone size [4,12]. Therefore, the present systematic review and meta-analysis compares RIRS, PCNL, and ESWL for renal calculus by stone size and analyzes treatment efficacy using various indexes like clinical aspects, retreatment rate, and complications. 

## 2. Materials and Methods

### 2.1. Inclusion Criteria

The selected studies were included based on the following set of inclusion criteria: (a) patient with calculus of the kidney; (b) comparative interventions between ESWL, PCNL, and RIRS to treat renal stones; and (c) reporting of at least one of the following outcome measures: stone-free rates (SFRs), complications, retreatment procedure, and auxiliary procedure. Search restrictions were set for English publication and human species. This report was prepared in compliance with the Preferred Reporting Items for Systematic Reviews and Meta-Analyses statement (accessible at http://www.prisma-statement.org/) [13]. The study was exempt from requiring approval of ethics committee or institutional review board because of systematic review and meta-analysis.

### 2.2. Search Strategy

A systemic review was performed to identify relevant studies that compared interventions used to treat calculus of the kidney using three English databases Ovid-Medline (1946–December 2017), Ovid-EMBASE (1974—December 2017), and the Cochrane Central Register of controlled Trials (1945–December 2017). We designed strategies that included Medical Subject Headings keywords such as “kidney calculi” “ureterolithiasis” “urolithiasis” “shock wave lithotripsy” “SWL” “percutaneous nephrolithotomy” “PCNL” “retrograde intrarenal surgery” “RIRS” “flexible ureteroscopy” “URSL” and combinations of search terms. Table 1 shows the characteristics of the included studies (Appendix A).

### 2.3. Study Selection and Data Extraction

To exclude irrelevant studies, two reviewers (SHL and CHK) independently screened the titles and abstracts of the articles and subsequently performed full-text screens of the potentially relevant articles. Two reviewers extracted the baseline demographics and clinical characteristics (i.e., age, sex) of the study participants onto a data extraction form and double-checked them.

### 2.4. Quality Assessment

A quality assessment was also independently performed by two reviewers (SHL and CHK) using the criteria provided by Scottish Intercollegiate Guidelines Network (SIGN). All discrepancies were resolved by discussion with a third reviewer. The SIGN checklist was used to assess the quality of randomized and nonrandomized studies with the advantage of a comprehensive parts assessment [14]. The tool consists of three areas: internal validity, overall assessment, and description [15]. The overall quality of the articles was indicated as 1++, 1+, and 1−.

### 2.5. Heterogeneity Tests

The heterogeneity of the included studies was examined using the Q statistic and Higgins’ I^2^ statistic [16]. Higgins’ I^2^ measures the percentage of total variation due to heterogeneity rather than chance across studies and was calculated as follows,
$$I^{2}=\frac{Q-\mathrm{df}}{Q}\times 100\%$$
in which “Q” is the Cochran’s heterogeneity statistic, and “df” is the degrees of freedom.

An I^2^ ≥ 50% is considered to represent substantial heterogeneity [17]. For the Q statistic, heterogeneity was deemed to significant at values of *p* < 0.10 [18]. If there was evidence of heterogeneity, the data were analyzed using a random-effects model. Studies in which positive results had been confirmed were assessed with a pooled specificity using 95% confidence intervals (CIs) [19,20].

### 2.6. Statistical Analysis

For dichotomous variables, the risk ratios (RRs) and weighted mean differences were calculated and reported with 95% CIs. As appropriate based on the degree of study heterogeneity, a random-effects model was applied to calculate summary measures. Meta-analyses were conducted using the Mantel–Haenszel and inverse variance methods, respectively. We conducted all meta-analyses using Review Manager version 5.3 (RevMan, Copenhagen: The Nordic Cochrane Center, The Cochrane Collaboration, 2013). All *p*-values were two-sided, and except for the test of discrepancy, a *p* < 0.05 was considered statistically significant.

## 3. Results

After full-text review, 32 articles were identified as relevant for this study. Five additional articles were included during our manual searches of the relevant bibliographies; therefore, 37 publications were ultimately included in the meta-analysis (Figure 1). 

### 3.1. Characteristics of Included Studies 

Table 1 shows the characteristics of the included studies. Twelve randomized controlled clinical trials (RCTs) and 25 observational studies met the eligibility criteria. The selected studies were published between 1991 and 2017. Nineteen studies were conducted in the Middle East [3,21,22,23,24,25,26,27,28,29,30,31,32,33,34,35,36,37,38], six in Europe [39,40,41,42,43,44], four in Asia [45,46,47,48], four in North America [49,50,51,52], two in South America [53,54], and two in Africa [55,56]. A total of 1460 PCNL cases, 1616 RIRS cases, and 2458 ESWL cases were compared in our meta-analysis. The included studies were divided by comparison type: six compared PCNL and ESWL [25,44,49,51,54,55], 13 compared PCNL and RIRS [22,26,27,29,30,31,33,37,38,39,47,48,52], 13 compared ESWL and RIRS [21,23,24,28,32,41,42,43,45,46,50,53,56], and five compared PCNL, ESWL, and RIRS [3,34,35,36,40]. Demographic characteristics such as mean age, sex ratio (male:female) were comparable among PCNL, RIRS, and ESWL study populations. (Table 1).

### 3.2. Quality Assessment and Publication Bias

The results of the quality assessment based on the Scottish Intercollegiate Guidelines Network checklist are shown in Table 1. Three studies [40,44,51] had a high risk of selection bias, indicating a quality of 1-. Concealment method and blinding were not reported. 

The funnel plot included in the meta-analysis is shown in Figure 2. There was little publication bias in all analyses (Figure 2).

### 3.3. Stone-Free Rate

Figure 3 shows a comparison of the SFRs of 37 studies. The definition between each study of SFRs is described in Table 1. We defined 4 mm sized or less as clinically insignificant residual fragments based on the definition of the included studies and previous reference studies [57,58]. Therefore, regardless of the location of the renal stones, 4 mm sized or less was defined as SFR and analyzed.

We conducted this meta-analysis according to intervention type and a sub-group analysis of stone size. The analysis was performed according to comparison: PCNL and ESWL, PCNL and RIRS, and ESWL and RIRS. PCNL provided a significantly higher SFR than RIRS (*p* < 0.001; RR = 1.14; 95% CI, 1.06–1.22; I^2^ = 69%) and ESWL (*p* < 0.001; RR = 0.69; 95% CI, 0.61–0.78; I^2^ = 81%). RIRS provided a significantly higher SFR than ESWL (*p* < 0.001; RR = 0.77; 95% CI, 0.67–0.88; I^2^ = 87%). A sub-group analysis was conducted based on the criterion of 2-cm stone diameter. A comparison of SFRs is shown in Figure 4. In the group of stones smaller than 2 cm, like in the primary analysis, PCNL provided a significantly higher SFR than RIRS (*p* = 0.002; RR = 1.08; 95% CI, 1.02–1.14; I^2^ = 0%), and ESWL (*p* < 0.001; RR = 0.76; 95% CI, 0.68–0.85; I^2^ = 63%). RIRS provided a significantly higher SFR than ESWL (*p* < 0.001; RR = 0.82; 95% CI, 0.73–0.92; I^2^ = 79%). In the group of stones larger than 2 cm, like in the primary analysis, PCNL provided a significantly higher SFR than RIRS (*p* < 0.001; RR = 1.23; 95% CI, 1.10–1.37; I^2^ = 74%), and ESWL (*p* < 0.001; RR = 0.72; 95% CI, 0.65–0.80), whereas no studies compared RIRS and ESWL.

### 3.4. Total Complications

A comparison of total complications is shown in Figure 5. Twenty-three studies reported on complications. We conducted a meta-analysis according to intervention type and a sub-group analysis by stone size. The analysis compared studies of PCNL and ESWL, PCNL and RIRS, and ESWL and RIRS. ESWL provided a lower total complications rate than PCNL (*p* < 0.001; RR, 0.28; 95% CI, 0.18–0.45; I^2^ = 64%). Whereas, no statistically significant difference was found between ESWL and RIRS (*p* = 0.38; RR, 0.87; 95% CI, 0.63–1.19; I^2^ = 16%). RIRS resulted in fewer total complications than PCNL (*p* = 0.02; RR, 1.41; 95% CI, 1.06–1.86; I^2^ = 15%).

Sub-group analysis was conducted on the criterion of 2-cm stone diameter. A comparison of SFRs is shown in Figure 6. In the group of stones smaller than 2 cm, like in the primary analysis, ESWL resulted in fewer complications than PCNL (*p* < 0.001; RR = 0.29; 95% CI, 0.16–0.52; I^2^ = 40%), whereas no statistically significant difference was found between ESWL and RIRS (*p* = 0.26; RR = 0.81; 95% CI, 0.56–1.17; I^2^ = 22%). RIRS resulted in fewer complications than PCNL (*p* = 0.009; RR = 2.05; 95% CI, 1.20–3.52; I^2^ = 0%). In the group of stones larger than 2 cm, unlike in the primary analysis, no significant difference was noted between PCNL and RIRS (*p* = 0.29; RR = 1.24; 95% CI, 0.84–1.83; I^2^ = 35%). No studies compared PCNL and ESWL or RIRS and ESWL.

### 3.5. Retreatment Procedure

A comparison of retreatment procedure is shown in Figure 7. Twenty-four studies reported on interventions as retreatment procedures. We conducted a meta-analysis according to intervention type and a sub-group analysis of stone size. The analysis was performed according to intervention type: PCNL versus SWL, PCNL versus RIRS, and SWL versus RIRS. ESWL required significantly more retreatment procedures than PCNL (*p* < 0.001; RR = 11.92; 95% CI, 3.06–46.53; I^2^ = 87%), and RIRS (*p* = 0.001; RR = 5.75; 95% CI, 1.99–16.60; I^2^ = 87%), while RIRS required significantly more retreatment procedures than PCNL (*p* = 0.02; RR = 0.29; 95% CI, 0.10–0.80; I^2^ = 66%).

A sub-group analysis was conducted of the criterion of 2-cm stone diameter. SFRs are compared in Figure 8. In the group of stones smaller than 2 cm, ESWL required significantly more retreatment procedures than PCNL (*p* < 0.001; RR = 43.33; 95% CI, 7.45–251.91; I^2^ = 39%), and RIRS (*p* = 0.001; RR = 6.25; 95% CI, 2.02–19.32; I^2^ = 89%), whereas there was no significant difference between PCNL and RIRS (*p* = 0.46; RR = 0.44; 95% CI, 0.05–3.61; I^2^ = 0%). In the group of stones larger than 2 cm, like in the primary analysis, ESWL required significantly more retreatment procedures than PCNL (*p* < 0.001; RR = 14.25; 95% CI, 7.50–27.07), and RIRS required significantly more retreatment procedures than PCNL (*p* = 0.01; RR = 0.22; 95% CI, 0.06–0.74, I^2^ = 75%). No studies compared RIRS and ESWL.

### 3.6. Auxiliary Procedure

Auxiliary procedure is compared in Figure 9. Twenty studies were included for auxiliary procedures. We conducted the meta-analysis of intervention types and then a sub-group analysis of stone size. The analysis was performed of the compared interventions: PCNL versus ESWL, PCNL versus RIRS, and ESWL versus RIRS. ESWL required more auxiliary procedures than PCNL (*p* = 0.05; RR = 2.39; 95% CI, 1.01–5.61; I^2^ = 64%), whereas there was no significant difference in use as an auxiliary procedure between ESWL and RIRS (*p* = 0.41; RR = 1.29; 95% CI, 0.70–2.40; I^2^ = 53%) or between PCNL and RIRS (*p* = 0.74; RR = 0.91; 95% CI, 0.50–1.63; I^2^ = 21%).

A sub-group analysis was conducted of the criterion of a 2-cm stone diameter. A comparison of SFRs is shown in Figure 10. In the group of stones smaller than 2 cm, like in the primary analysis, there was no significant difference in use as an auxiliary procedure between ESWL and PCNL (*p* = F 0.75; RR = 1.45; 95% CI, 0.15–13.63; I^2^ = 81%), between ESWL and RIRS (*p* = 0.56; RR = 1.2; 95% CI, 0.65–2.23; I2 = 54%), and between PCNL and RIRS (*p* = 0.70; RR = 0.84; 95% CI, 0.34–2.07; I^2^ = 0%). In the group of stones larger than 2 cm, there was no significant difference in use as an auxiliary procedure between PCNL and RIRS (*p* = 0.64; RR = 1.37; 95% CI, 0.37–5.03; I^2^ = 66%). Unlike in the primary analysis, ESWL required a significantly more auxiliary procedure than PCNL (*p* < 0.001; RR = 5.34; 95% CI, 2.59–11.04). No study compared RIRS and ESWL.

We summarize all the analyzed results in Table 2. In addition, Table 2 also includes sub-analysis on the lower pole stone.

## 4. Discussion

The use of minimally invasive techniques, such as PCNL, ESWL, and RIRS has increased dramatically over the last 30 years through the sustained high incidence and recurrence of renal stones [59]. New procedures are being introduced with the combination of instruments and technology. Since Fernstrom and Johansson introduced PCNL for the first time in 1976 as a surgical treatment for patients with large and complex renal stones, PCNL has been considered a standard surgery for stones larger than 2 cm [60,61]. As advances in instruments and technology develop, PCNL has evolved from tubeless PCNL to supine PCNL to mini PCNL [62,63,64]. Next, ESWL was first reported in 1984 after Chaussy and colleagues performed SWL on 852 patients [65]. SWL is a relatively noninvasive procedure that has been used as a first treatment choice for small renal stones less than 2 cm not within the lower pole of the kidney. Finally, RIRS has progressed rapidly since the 1990 s, when the holmium:yttrium aluminum garnet laser system was introduced. RIRS become popular with the development of the more durable models such as Flex-X from Karl Storz Endoskope, Tuttlingen, Germany and URF-P from Olympus, Tokyo, Japan. Also, the recently introduced compact aperture digital video scope and disposable video scope contributed to becoming more popular of RIRS [66,67]. RIRS is constantly evolving and has also recently been used to treat renal sinus cysts besides stones [68]. 

Three procedures are now widely used by urologists in the treatment of renal stones. Guidelines recommend certain procedures according to stone location and size, but patient- and doctor-related factors commonly result in other choices. The EAU guidelines [6] suggest the use of ESWL or RIRS for the primary treatment of renal stones smaller than 2 cm and PCNL for the primary treatment of stones larger than 2 cm. Perhaps SFR is one of the first points to consider when choosing among treatments for renal stones, as each has its own advantages and disadvantages. Also, the complication and auxiliary procedure rates may be important factors. Therefore, our meta-analysis aimed to help urologists make better treatment decisions by providing them a comparison of these three procedures and a sub-analysis by stone size (smaller versus greater than 2 cm). 

In general, although PCNL is the most effective interventional therapy with the highest SFR, careful patient selection is required due to its highly invasive nature [4,67]. In our results, PCNL also showed the best SFR for overall renal stone treatment, statistically significantly compared to the other two procedures. In a sub-analysis according to stone size, RIRS and PCNL showed a similar SFR for stones smaller than 2 cm, whereas PCNL showed better results than RIRS for stones greater than 2 cm. As a result, PCNL had the lowest retreatment rate. The complication rate, the largest disadvantage of PCNL, was relatively higher than those of ESWL and RIRS when stone size was not considered. However, complication rates did not differ between PCNL and RIRS in cases of stones larger than 2 cm. This is the most important finding of our study since it demonstrates that there is no need to choose ESWL or RIRS simply because of the high complication rate of PCNL in renal stones greater than 2 cm. Recent reports have highlighted the advantage of the decreased invasiveness of mini PCNL and ultra-mini PCNL [69,70]. We also think that improvements to mini PCNL and ultra PCNL through the development of PCNL technology have contributed greatly to reducing this complication rate. 

There are several limitations to our study. First, sub-analysis according to location was partially limited due to limitations in providing information according to stone location for each study. Such an analysis may have led to different outcomes because the recommended treatments vary depending on renal stone location. Next, sub-analyses of mini PCNL, ultra PCNL, and conventional PCNL could not be performed. We think that further studies will need to study the utility of PCNL types through these analyses. Finally, some degree of publication bias was unavoidable, and the non-RCTs may involve additional selection bias. 

Despite these limitations, our study still has its advantages. As mentioned earlier, PCNL has already been presented as a first choice through the EAU guidelines in large renal stones, but many urologists are concerned about PCNL-related complications. However, our meta-analysis including all published studies showed that the complication rate of PCNL is not high in cases of stones larger than 2 cm. Therefore, PCNL is a safe and effective treatment, not a hazardous procedure, compared with other procedures for large renal stones. We believe that this finding can guide the safety and efficacy of PCNL for urologists treating large renal stones. This study may be a good source of information for patients regarding renal stone therapy.

## 5. Conclusions

In our meta-analysis, among procedures for renal stones, PCNL had the highest SFR and lower auxiliary procedure and retreatment rates than ESWL or RIRS. No statistically significant difference was seen versus RIRS in complications regarding stones larger than 2 cm. Therefore, we suggest that PCNL is a safe and effective treatment, particularly for large renal stones.

## Figures and Tables

**Figure 1 medicina-57-00026-f001:**
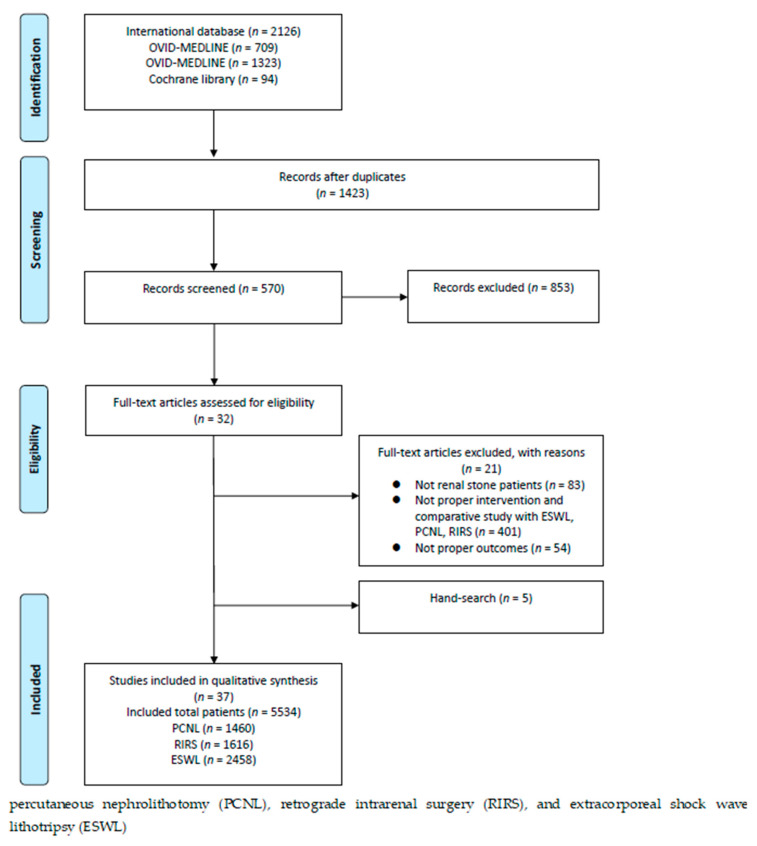
Study flow chart. Thirty-seven studies were included in the qualitative analysis.

**Figure 2 medicina-57-00026-f002:**
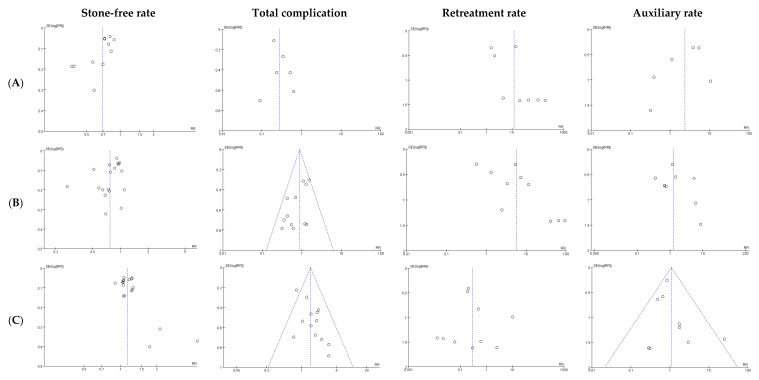
Funnel plot of the meta-analysis. (**A**) ESWL vs. PCNL; (**B**) ESWL vs. RIRS; (**C**) RIRS vs. PCNL. These funnel plots indicate the presence of little publication bias in all studies.

**Figure 3 medicina-57-00026-f003:**
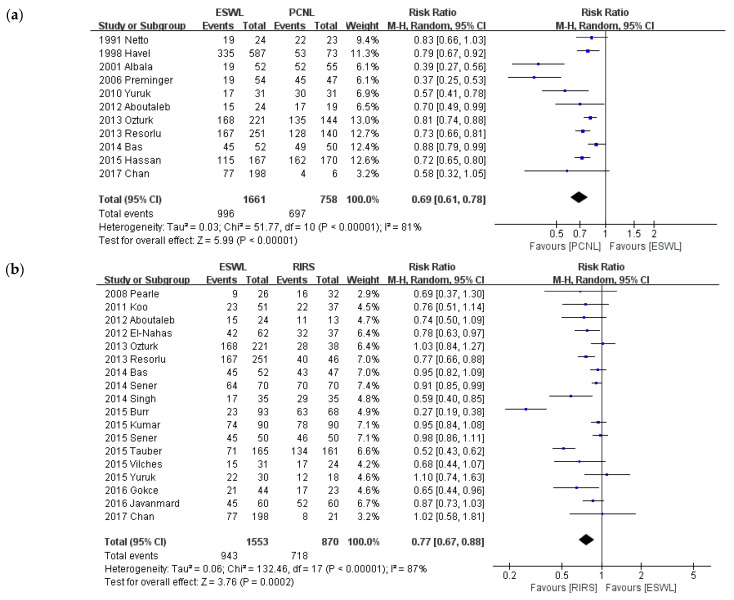
Meta-analysis of Stone-free rate: (**a**) ESWL versus PCNL; (**b**) ESWL versus RIRS; and (**c**) PCNL versus RIRS. Mantel-Haenszel (M-H), 95% confidence intervals (CI).

**Figure 4 medicina-57-00026-f004:**
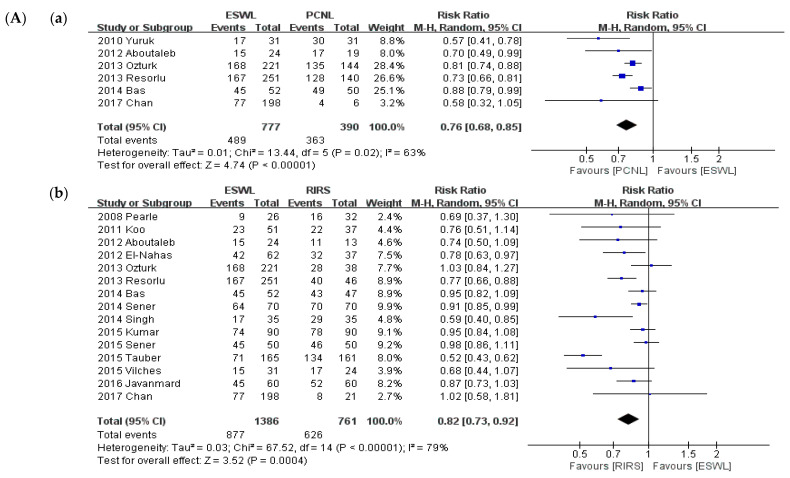
Meta-analysis (**A**). Stone-free rate (stone size ≤ 2 cm): (**a**) ESWL vs. PCNL; (**b**) ESWL vs. RIRS; (**c**) and PCNL vs. RIRS. Meta-analysis (**B**). Stone-free rate (stone size >2 cm): (**a**) ESWL vs. PCNL; and (**b**) PCNL vs. RIRS.

**Figure 5 medicina-57-00026-f005:**
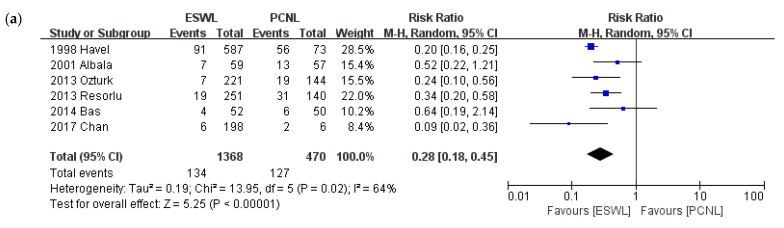
Meta-analysis of total complications: (**a**) ESWL vs. PCNL; (**b**) ESWL vs. RIRS; and (**c**) PCNL vs. RIRS.

**Figure 6 medicina-57-00026-f006:**
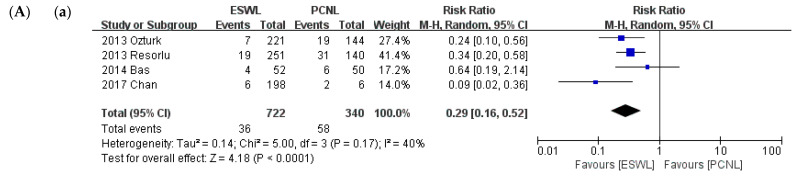
Meta-analysis (**A**). Total complications (stone size ≤ 2 cm): (**a**) ESWL vs. PCNL; (**b**) ESWL vs. RIRS; and (**c**) PCNL vs. RIRS. Meta-analysis (**B**). Total complications (stone size > 2 cm): (**a**) PCNL vs. RIRS.

**Figure 7 medicina-57-00026-f007:**
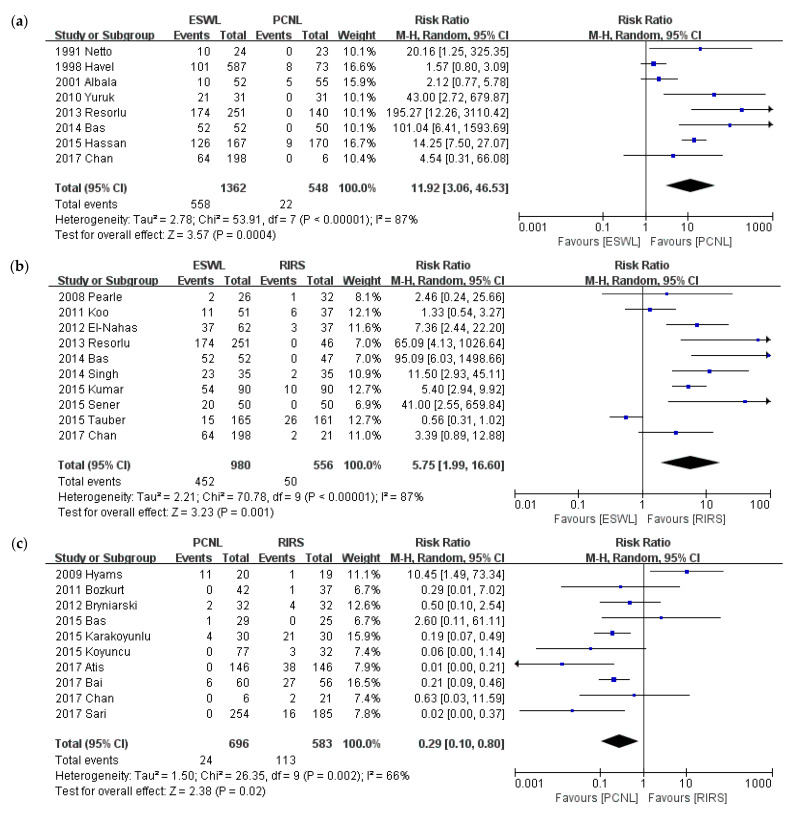
Meta-analysis of retreatment procedure: (**a**) ESWL vs. PCNL; (**b**) ESWL vs. RIRS; (**c**) PCNL vs. RIRS.

**Figure 8 medicina-57-00026-f008:**
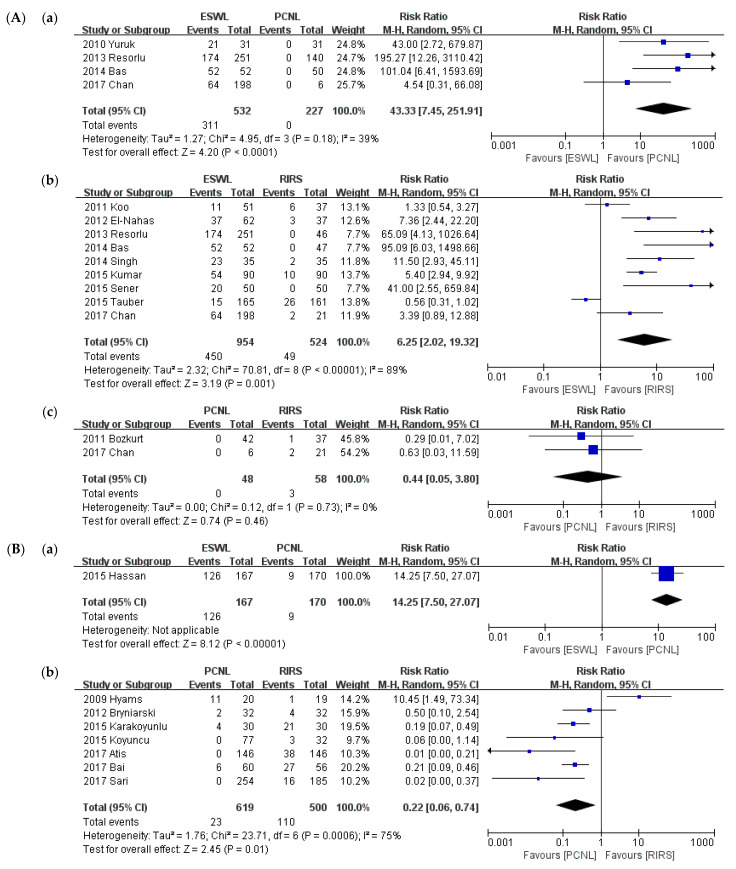
Meta-analysis (**A**). Retreatment procedure (stone size ≤ 2 cm): (**a**) ESWL vs. PCNL; (**b**) ESWL vs. RIRS; and (**c**) PCNL vs. RIRS. Meta-analysis (**B**). Retreatment procedure (stone size > 2 cm): (**a**) ESWL vs. PCNL; and (**b**) PCNL vs. RIRS.

**Figure 9 medicina-57-00026-f009:**
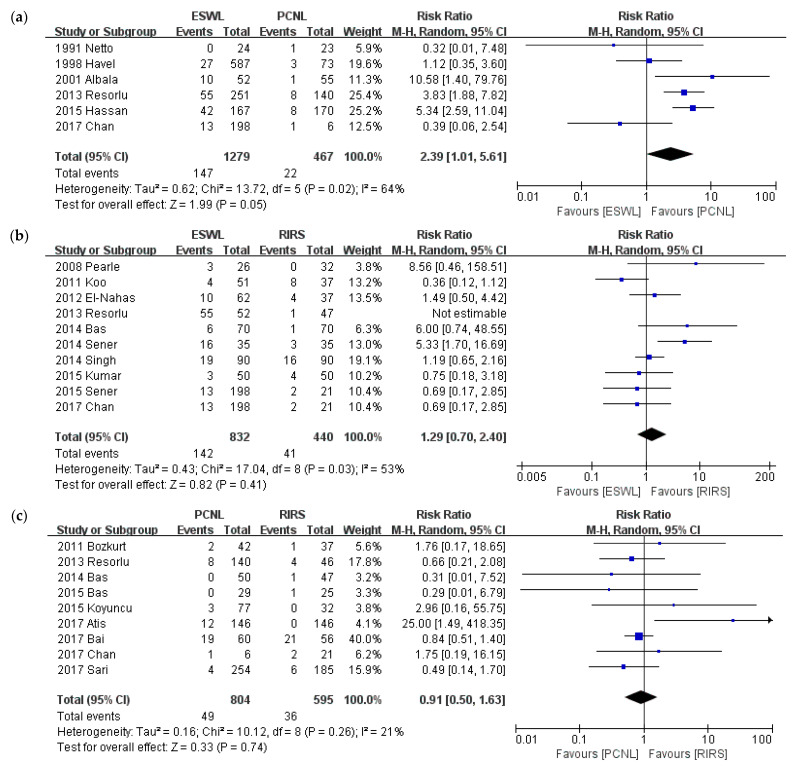
Meta-analysis of auxiliary procedure: (**a**) ESWL vs. PCNL; (**b**) ESWL vs. RIRS; and (**c**) PCNL vs. RIRS.

**Figure 10 medicina-57-00026-f010:**
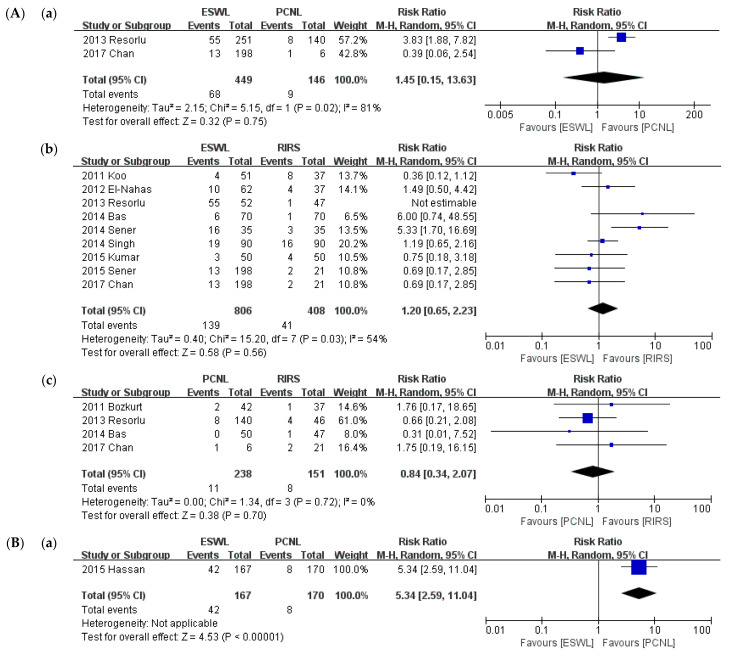
Meta-analysis (**A**). Auxiliary procedure (stone size ≤ 2 cm): (**a**) ESWL vs. PCNL; (**b**) ESWL vs. RIRS; and (**c**) PCNL vs. RIRS. Meta-analysis (**B**). Auxiliary procedure (stone size > 2 cm): (**a**) ESWL vs. PCNL; and (**b**) PCNL vs. RIRS.

**Table 1 medicina-57-00026-t001:** Characteristics of included studies.

Author Year	Institution (Country)	Study Design	Study Quality	Inclusion Criteria	Definition of Stone-Free Rate	Sample Size			Age *			Male:Female		
						PCNL	RIRS	ESWL	PCNL	RIRS	ESWL	PCNL	RIRS	ESWL
Javanmard [21] 2016	Multiple institutions (Iran)	RCT	++	0.6–2 cm	Residual stone fragments ≤3 mm	-	60	60		32.4 ± 7.8	31.3 ± 6.5		37:23	39:21
Karakoyunlu [22] 2015	Diskapi Yildirim Beyazit Training & Research Hospital (Turkey)	RCT	+	>2 cm	Radiological absence of stone, or residual stone fragments <4 mm	30	30	-	45.8 ± 14.1	48.4 ± 15.5	-	18:12	16:14	-
Kumar [45] 2015	Safdarjang Hospital (India)	RCT	++	≤2 cm	Radiological absence of stone, asymptomatic patients with residual stone fragments <3 mm	-	90	90	-	35.6 ± 2.1	37.7 ± 2.4	-	46:44	44:46
Sener [23] 2015	Numune Education and Research Hospital (Turkey)	RCT	+	<1 cm, lower pole, asymptomatic	Residual stone fragments <3 mm	-	50	50		36.84 ± 11.70	34.5 ± 11.04		35:15	37:13
Vilches [53] 2015	Clinical Hospital University of Chile (Chile)	RCT	+	<1.5 cm, lower pole	Radiological absence of stone	-	24	31		43.7 ± 9.2	45.6 ± 13.7		15:9	18:13
Singh [46] 2014	King George Medical University (India)	RCT	+	1–2 cm, inferior calyx	Radiological absence of stone	-	35	35		37.65 ± 11.8	34.5 ± 13.07		22:13	20:15
Sener [24] 2014	Numune Education and Research Hospital (Turkey)	RCT	+	<1 cm, lower pole	Residual stone fragments <3 mm	-	70	70	-	45.4 ± 6.4	42.9 ± 5.6	-	41:29	31:29
Bryniarski [39] 2012	Silesian Medical University (Poland)	RCT	+	≥2 cm	Radiological absence of stone	32	32	-	51.8 ± 11.8	53.4 ± 12.4	-	16:16	15:17	-
Yuruk [25] 2010	Haseki Teaching & Research Hospital (Turkey)	RCT	+	≤2 cm, lower pole, asymptomatic	Radiological absence of stone	31	-	31	44.1 ± 12.3	-	44.5 ± 9.4	15:16	-	16:15
Preminger [49] 2006	Duke University Medical Center (US)	RCT	+	≤3 cm, lower pole	Radiological absence of stone	47	-	54						
Pearle [50] 2005	Multiple institutions (US)	RCT	+	<1 cm, lower pole	Radiological absence of stone	-	32	26	-	49.3 ± 14.2	52.5 ± 12.3	-	17:18	19:13
Albala [51] 2001	Methodist Hospital (US)	RCT	-	≤3 cm, lower pole, symptomatic	Residual stone fragments <3 mm	55	-	52						
Chan [40] 2017	Western General Hospital (Scotland)	Prospective	-	1–2 cm, lower pole	Residual stone fragments ≤ 3 mm	6	21	198	55.2 ± 20.3	62.2 ± 15	54.1 ± 13.3	50:50	52:48	73:27
Ozayar [26] 2016	Multiple institutions (Turkey)	Prospective	+	<2 cm, lower pole	Not specific	30	26	-	47.8 ± 11.4	42.5 ± 10.9		20:10	14:12	
Atis [27] 2017	Multiple institutions (Turkey)	Retrospective	++	2–4 cm	Radiological absence of stone	146	146	-	46.33 ± 12.34	47.23 ± 15.16		104:42	98:48	
Bai [47] 2017	Sichuan University (China)	Retrospective	+	>2 cm, solitary kidney	Radiological absence of stone	60	56	-	52.22 ± 10.56	48.84 ± 11.27	-	44:16	37:19	-
Sari [29] 2017	Sarikamis State Hospital (Turkey)	Retrospective	+	≥2 cm	Asymptomatic patients with residual stone fragments <3 mm	254	185	-	46.88 ± 14.35	48.04 ± 14.09		155:99	111:74	
Gokce [28] 2016	Ankara University (Turkey)	Retrospective	+	any size, horseshoe kidney	Residual stone fragments <3 mm	-	23	44	-	44.2 ± 9.9	42.8 ± 8.4	-	18:5	32:12
Hassan [55] 2015	Multiple institutions (Egypt)	Retrospective	+	2–3 cm	Residual stone fragments ≤ 4 mm	170-	-	167	50.9 ± 12.4	-	47.7 ± 11.7	80:90	-	107:60
Bas [30] 2015	Abdurrahman Yurtaslan hospital (Turkey)	Retrospective	+	any size, symptomatic	Radiological absence of stone	29	25	-	45.10 ± 10.53	36.28 ± 10.43	-	14:15	12:13	-
Burr [41] 2015	Southampton University Hospital (UK)	Retrospective	+	any size, lower pole	Residual stone fragments ≤ 3 mm	-	68	93		54 ± 16.6	54 ± 14.6		39:29	60:33
Koyuncu [31] 2015	Multiple institutions (Turkey)	Retrospective	+	≥2 cm, lower pole	Radiological absence of stone	77	32	-	38.7 ± 13.6	40.7 ± 15.8		45:32	20:12	
Jung [48] 2015	Multiple institutions (Korea)	Retrospective	+	1.5–3 cm, lower pole	Asymptomatic patients with residual stone fragments <3 mm	44	44	-	56.1 ± 13.2	53.8 ± 13.4		52:35	29:15	
Tauber [42] 2015	Multiple institutions (Austria)	Retrospective	+	≤1.5 cm	Radiological absence of stone	-	161	165		53.9	52.0		37.3:62.7	36.4:63.6
Yuruk [32] 2015	Haseki Training and Research Hospital (Turkey)	Retrospective	++	any size, solitary kidney	Not specific	-	18	30		47.1 ± 13.8	51.7 ± 17.8		9:9	22:8
Zengin [33] 2015	Bozok University (Turkey)	Retrospective	+	2–3 cm	Residual stone fragments < 2 mm	74	80	-	45.6	48.3		34:40	38:42	
Bas [3] 2014	Abdurrahman Yurtaslan hospital (Turkey)	Retrospective	+	1–2 cm	Radiological absence of stone	50	47	42	45.54 ± 13.1	47.2 ± 14.2	46.4 ± 15.1	58:42	63.82:36.17	53.84:46.15
Resorlu [34] 2013	Multiple institutions (Turkey)	Retrospective	+	1–2 cm	Radiological absence of stone, or asymptomatic patients with residual stone < 3 mm	140	46	251	36.4 ± 19.7	29.6 ± 20.3	30.8 ± 15.9	72:68	24:22	175:76
Ozturk [35] 2013	Yıldırım Beyazit Education and Research Hospital (Turkey)	Retrospective	+	1–2 cm, lower pole	Radiological absence of stone, or residual stone fragments < 3 mm	144	38	221	41.1	52	44.2	88:56	22:16	123:98
Aboutaleb [36] 2012	Farwaniya Hospital (Kuwait)	Retrospective	+	1–2 cm, lower pole	Residual stone fragments <3 mm	19	13	24	45.3 ± 14.3	47.2 ± 15.2	53.2 ± 19	14:5	7:6	19:5
El-nahas [56] 2012	Mansoura University (Egypt)	Retrospective	+	1–2 cm, lower pole	Residual stone fragments <4 mm	-	37	62		47.8 ± 10.7	45.4 ± 11.3		26:11	41:21
Akman [37] 2011	Haseki Training & Research Hospital (Turkey)	Retrospective	+	2–4 cm	Radiological absence of stone	34	34	-	44.8 ± 17.1	44.5 ± 16.5		47.1:52.9	52.9:47.1	
Bozkurt [38] 2011	Kecioren Research & Training Hospital (Turkey)	Retrospective	+	1.5–2 cm	Residual stone fragments <3 mm	42	37	-	47.4 ± 15.5	41.2 ± 13.6		25:17	21:16	
Koo [43] 2011	Craigavon Area Hospital (UK)	Retrospective	+	≤2 cm, lower pole	Radiological absence of stone, or residual stone fragments < 3 mm	-	37	51		56.6 ± 15.9	51.2 ± 14.9		22:15	35:16
Hyams [52] 2009	New York University (US)	Retrospective	+	2–3 cm	Residual stone fragments <4 mm	20	19	-	48	56		11:9	5:14	
Havel [44] 1998	Strasbourg University (France)	Retrospective	-	any size, lower pole	Radiological absence of stone	73	-	587	50	-	48			
Netto [54] 1991	Israelita Albert Einstiein Hospital (Brazil)	Retrospective	+	≤3 cm	Radiological absence of stone	23	-	24	44		48	13:10		14:10

PCNL, percutaneous nephrolithotomy; RCT, randomized controlled trial; RIRS, retrograde intrarenal surgery; SWL, extracorporeal shock wave lithotripsy. * Age is shown as mean ± SD (standard deviation). The overall quality of the articles was indicated as 1++, 1+, and 1− by Scottish Intercollegiate Guidelines Network (SIGN).

**Table 2 medicina-57-00026-t002:** Summary of results.

	**Outcome**	**No. of Studies**	**RR (95% CI)**	***p* Value**	**I^2^ (%)**
ESWL vs. PCNL	Stone-free rate	11	0.69 (0.61–0.78)	<0.00001	81
	stone size ≤ 2 cm	6	0.76 (0.68–0.85)	<0.00001	63
	stone size > 2 cm	1	0.72 (0.65–0.80)	<0.00001	-
	Lower pole stone	6	0.58 (0.44–0.76)	0.001	88
	Total complication	6	0.28 (0.18–0.45)	<0.00001	64
	stone size ≤ 2 cm	4	0.29 (0.16–0.52)	<0.0001	40
	stone size > 2 cm	0	-	-	-
	Lower pole stone	4	0.24 (0.14–0.41)	<0.00001	57
	Retreatment procedure	8	11.92 (3.06–46.53)	0.0004	87
	stone size ≤ 2 cm	4	43.33 (7.45–251.91)	<0.0001	39
	stone size > 2 cm	1	14.25 (7.50–27.07)	<0.00001	-
	Lower pole stone	4	2.91 (0.99–8.56)	0.05	58
	Auxiliary procedure	6	2.39 (1.01–5.61)	0.05	64
	stone size ≤ 2 cm	2	1.45 (0.15–13.63)	0.75	81
	stone size > 2 cm	1	5.34 (2.59–11.04)	<0.00001	-
	Lower pole stone	3	1.56 (0.27–8.93)	0.62	70
	**Outcome**	**No. of Studies**	**RR (95% CI)**	***p* Value**	**I^2^ (%)**
ESWL vs. RIRS	Stone-free rate	18	0.77 (0.67–0.88)	0.0002	87
	stone size ≤ 2 cm	15	0.82 (0.73–0.92)	0.0004	79
	stone size > 2 cm	-	-	-	-
	Lower pole stone	10	0.76 (0.60–0.97)	0.03	92
	Total complication	12	0.87 (0.63–1.19)	0.38	16
	stone size ≤ 2 cm	10	0.81 (0.56–1.17)	0.26	22
	stone size > 2 cm	-	-	-	-
	Lower pole stone	6	0.49 (0.29–0.83)	0.008	0
	Retreatment procedure	10	5.75 (1.99–16.60)	0.001	87
	stone size ≤ 2 cm	9	6.25 (2.02–19.32)	0.001	89
	stone size > 2 cm	-	-	-	-
	Lower pole stone		3.96 (1.35–11.60)	0.01	63
	Auxiliary procedure	10	1.29 (0.70–2.40)	0.41	53
	stone size ≤ 2 cm	9	1.2 (0.65–2.23)	0.56	54
	stone size > 2 cm	-	-		-
	Lower pole stone	5	1.46 (0.48–4.48)	0.50	70
	**Outcome**	**No. of studies**	**RR (95% CI)**	** *p* ** **value**	**I^2^ (%)**
PCNL vs. RIRS	Stone-free rate	18	1.14 (1.06–1.22)	0.0003	69
	stone size ≤ 2 cm	7	1.08 (1.02–1.14)	0.01	0
	stone size > 2 cm	9	1.23 (1.10–1.37)	0.0003	74
	Lower pole stone	5	1.34 (0.95–1.88)	0.10	89
	Total complication	13	1.41 (1.06–1.86)	0.02	15
	stone size ≤ 2 cm	5	2.05 (1.20–3.52)	0.009	0
	stone size > 2 cm	6	1.24 (0.84–1.83)	0.29	35
	Lower pole stone	4	1.99 (0.91–4.37)	0.09	10
	Retreatment procedure	10	0.29 (0.10–0.80)	0.02	66
	stone size ≤ 2 cm	2	0.44 (0.05–3.80)	0.46	0
	stone size > 2 cm	7	0.22 (0.06–0.74)	0.01	75
	Lower pole stone	2	0.20 (0.02–1.97)	0.17	20
	Auxiliary procedure	9	0.91 (0.50–1.63)	0.74	21
	stone size ≤ 2 cm	4	0.84 (0.34–2.07)	0.70	0
	stone size > 2 cm	4	1.37 (0.37–5.03)	0.64	66
	Lower pole stone	2	2.12 (0.36–12.47)	0.41	0

PNL, percutaneous nephrolithotomy; RIRS, retrograde intrarenal surgery; SWL, extracorporeal shock wave lithotripsy RR, risk ratio; CI, confidence intervals.

## Data Availability

The data presented in this study are available in the article.

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
