# Peer review of "Effectiveness of Percutaneous Nephrolithotomy, Retrograde Intrarenal Surgery, and Extracorporeal Shock Wave Lithotripsy for Treatment of Renal Stones: A Systematic Review and Meta-Analysis"

_medicina, 2020, doi:10.3390/medicina57010026_

Round 1

Reviewer 1 Report

The manuscript by Kim et al is a systematic review and meta-analysis about the more used treatments for renal stones. The authors have conducted an accurate analysis of data and, given the complexity involved, they have produced many positive and welcome outcomes. As stated by the authors in their discussion, “This study may be a good source of information for patients regarding renal stone therapy”.

Notwithstanding the foregoing, in my opinion, this study “might be” a good source of information for patients regarding renal stone therapy. In fact, despite the considerable effort put by the authors, this manuscript requires thorough revision, and it is incomplete.

Majors

First, chapter 3.1 is concluded with the sentence “The included studies were divided by comparison type: six”. Where is the last part of this chapter?

Second: chapter 3.2 is inexistent

Line 70: “Table 1 shows the characteristics of the included studies (Supplement Table 1)”. ?? I did not find any Supplement Table 1

This paper would benefit from some closer proofreading. It includes many linguistic errors (e.g. agreement of verbs) that at times make it difficult to follow. Besides, there are a few sentences that need rephrasing for clarity. For instance, in line 40 develop should be developed, in line 48-49 you should avoid repetitions as “compared” and compares….Please check the whole text.

Minor points

Line 23: “Auxiliary” should be “auxiliary”. Capital letter is unnecessary

Key words: Please delete “systematic review” and “meta-analysis”

Line 43: “Guideline” should be “guidelines”

Line 48: “reviews” instead of review

Line 67: Cochrane Central Register of controlled Trials. How many years?

Line 91: p < .10. What is this?

Line 250: guidelines

Line 295: This is very strange. The authors stated “no external funding” but, then, they thank the Korea Health Technology R&D Project of the Korea Health Industry Development Institute (KHIDI), funded by the Ministry of Health & Welfare, Republic of Korea. Simply, I do not understand it

Author Response

Dear Reviewer

Thank you for your thoroughly reviewing our manuscript (medicina-1024184) entitled “Effectiveness of Percutaneous Nephrolithotomy, Retrograde Intrarenal Surgery, and Extracorporeal Shock Wave Lithotripsy for Treatment of Renal Stones: A Systematic Review and Meta-Analysis” Also, we are grateful for the chance to revise our manuscript. Our manuscript has been carefully revised according to the reviewers’ comments. Please find our responses to the reviewer’ comments beginning on the next page.

We hope that our revised paper is acceptable for publication in Medicina, and we look forward to receiving your final decision.

Thanks, again.

Sincerely,

*Co-corresponding authors: Joo Yong Lee, MD, PhD, and Seon Heui Lee, PhD

Joo Yong Lee, MD, PhD

Department of Urology, Severance Hospital, Urological Science Institute, Yonsei University College of Medicine, 50-1 Yonsei-ro, Seodaemun-gu, Seoul 03722, Korea

Center of Evidence Based Medicine, Institute of Convergence Science, Yonsei University, Seoul 03722, Korea;

Tel: +82-2-2228-2320; Fax: +82-2-312-2538; E-mail: [email protected]

Seon Hui Lee, PhD

Department of Nursing Science, College of Nursing, Gachon University, 191 Hambangmoe-ro, Yeonsu-gu, Incheon 21936, Korea

Tel: +82-32-820-4235; Fax: +82-32-820-4201; E-mail: [email protected]

#Reviewer 1

The manuscript by Kim et al is a systematic review and meta-analysis about the more used treatments for renal stones. The authors have conducted an accurate analysis of data and, given the complexity involved, they have produced many positive and welcome outcomes. As stated by the authors in their discussion, “This study may be a good source of information for patients regarding renal stone therapy”.

Notwithstanding the foregoing, in my opinion, this study “might be” a good source of information for patients regarding renal stone therapy. In fact, despite the considerable effort put by the authors, this manuscript requires thorough revision, and it is incomplete.

[Answers]

Thank you for your comment. We also agree with your feedback.

Thank you for your careful consideration.

[Major points]

Q1. First, chapter 3.1 is concluded with the sentence “The included studies were divided by comparison type: six”. Where is the last part of this chapter?

A1. Sorry. We have written for the rest of the chapter.

3.1. Characteristics of Included Studies

Table 1 shows the characteristics of the included studies. Twelve randomized controlled clinical trials (RCTs) and 25 observational studies met the eligibility criteria. The selected studies were published between 1991 and 2017. Nineteen studies were conducted in the Middle East [3,21-38], six in Europe [39-44], four in Asia [45-48], four in North America [49-52], two in South America [53,54], and two in Africa [55,56]. A total of 1,460 PCNL cases, 1,616 RIRS cases, and 2,458 ESWL cases were compared in our meta-analysis. The included studies were divided by comparison type: six compared PCNL and ESWL [25,44,49,51,54,55], 13 compared PCNL and RIRS [22,26,27,29-31,33,37-39,47,48,52], 13 compared ESWL and RIRS [21,23,24,28,32,41-43,45,46,50,53,56], and five compared PCNL, ESWL, and RIRS [3,34-36,40]. Demographic characteristics such as mean age, sex ratio (male:female) were comparable among PCNL, RIRS, and ESWL study populations. (Table 1)

Q2. Second: chapter 3.2 is inexistent

A2. Sorry. Chapter 3.2 was written with the quality assessment and publication bias chapter.

3.2. Quality assessment and publication bias

The results of the quality assessment based on the Scottish Intercollegiate Guidelines Network checklist are shown in Table 1. Three studies [40,44,51] had a high risk of selection bias, indicating a quality of 1-. Concealment method and blinding were not reported.

The funnel plot included in the meta-analysis is shown in Fig. 2. There was little publication bias in all analyses (Fig. 2).

Q3. Line 70: “Table 1 shows the characteristics of the included studies (Supplement Table 1)”. ?? I did not find any Supplement Table 1

A3. Supplement Table1 was written about the search results. We added it at the end of the text.

Q4. This paper would benefit from some closer proofreading. It includes many linguistic errors (e.g. agreement of verbs) that at times make it difficult to follow. Besides, there are a few sentences that need rephrasing for clarity. For instance, in line 40 develop should be developed, in line 48-49 you should avoid repetitions as “compared” and compares….Please check the whole text.

A4. We did proofreading through native speakers before submission, but it seems to be insufficient. The revised version was recalibrated. Thank you.

[Minor points]

Line 23: “Auxiliary” should be “auxiliary”. Capital letter is unnecessary

Key words: Please delete “systematic review” and “meta-analysis”

Line 43: “Guideline” should be “guidelines”

Line 48: “reviews” instead of review

Line 67: Cochrane Central Register of controlled Trials. How many years?

Line 91: p < .10. What is this?

Line 250: guidelines

Line 295: This is very strange. The authors stated “no external funding” but, then, they thank the Korea Health Technology R&D Project of the Korea Health Industry Development Institute (KHIDI), funded by the Ministry of Health & Welfare, Republic of Korea. Simply, I do not understand it

[Answer]

 Thank you for your attentive comments. We have corrected all the points mentioned. In addition, this work was supported by the Korea Health Technology R&D Project of the Korea Health Industry Development Institute (KHIDI), funded by the Ministry of Health & Welfare, Republic of Korea. But, the funding body had no influence in the design of the study and collection, analysis, and interpretation of data and in writing the manuscript.

Reviewer 2 Report

The authors have provided a useful systematic review and meta-analysis in the area of treatments for renal stone. Specifically, the authors focus on percutaneous nephrolithotomy, retrograde intrarenal surgery, and extracorporeal shock wave lithotripsy, and perform their analysis with respect to effectiveness, in addition to other criteria such as retreatment, complications, and stone size. In this way, this review is not only more comprehensive than similar counterparts in comparing three techniques, but the scope is greater in the meta-analyses being performed also.

Overall, this is a well written, comprehensive account of the area, with the authors highlighting the limitations of the study themselves, and the subsequent impact of such. I believe this will prove a useful resource for those interested in this particular area.

My only critique would be that the graphs could be of higher quality/formatting. At times the data is difficult to interpret given the formatting, and if the authors can improve on this for Figures 3-10 that would improve the manuscript. 

Author Response

Dear Reviewer

Thank you for your thoroughly reviewing our manuscript (medicina-1024184) entitled “Effectiveness of Percutaneous Nephrolithotomy, Retrograde Intrarenal Surgery, and Extracorporeal Shock Wave Lithotripsy for Treatment of Renal Stones: A Systematic Review and Meta-Analysis” Also, we are grateful for the chance to revise our manuscript. Our manuscript has been carefully revised according to the reviewers’ comments. Please find our responses to the reviewer’ comments beginning on the next page.

We hope that our revised paper is acceptable for publication in Medicina, and we look forward to receiving your final decision.

Thanks, again.

Sincerely,

*Co-corresponding authors: Joo Yong Lee, MD, PhD, and Seon Heui Lee, PhD

Joo Yong Lee, MD, PhD

Department of Urology, Severance Hospital, Urological Science Institute, Yonsei University College of Medicine, 50-1 Yonsei-ro, Seodaemun-gu, Seoul 03722, Korea

Center of Evidence Based Medicine, Institute of Convergence Science, Yonsei University, Seoul 03722, Korea;

Tel: +82-2-2228-2320; Fax: +82-2-312-2538; E-mail: [email protected]

Seon Hui Lee, PhD

Department of Nursing Science, College of Nursing, Gachon University, 191 Hambangmoe-ro, Yeonsu-gu, Incheon 21936, Korea

Tel: +82-32-820-4235; Fax: +82-32-820-4201; E-mail: [email protected]

# Reviewer 2

The authors have provided a useful systematic review and meta-analysis in the area of treatments for renal stone. Specifically, the authors focus on percutaneous nephrolithotomy, retrograde intrarenal surgery, and extracorporeal shock wave lithotripsy, and perform their analysis with respect to effectiveness, in addition to other criteria such as retreatment, complications, and stone size. In this way, this review is not only more comprehensive than similar counterparts in comparing three techniques, but the scope is greater in the meta-analyses being performed also.

Overall, this is a well written, comprehensive account of the area, with the authors highlighting the limitations of the study themselves, and the subsequent impact of such. I believe this will prove a useful resource for those interested in this particular area.

My only critique would be that the graphs could be of higher quality/formatting. At times the data is difficult to interpret given the formatting, and if the authors can improve on this for Figures 3-10 that would improve the manuscript. 

[Answers]

Thank you for your comment.

Thank you for your careful consideration.

We also agree with your feedback. We will try to publish higher quality figures by providing original pictures with good quality if possible.

Reviewer 3 Report

This is a meta-analysis based on 37 publications that compared RIRS (retrograde intrarenal surgery), PCNL (percutaneous nephrolithotomy), and ESWL (extracorporeal shock wave lithotripsy) treatments for renal calculus by stone size and analyses treatment efficacy using various indexes like clinical aspects, retreatment rate, and complications.

From their analysis, the authors found that RIRS requires significantly more retreatment procedures than PCLN for > 2 cm stones. The complication was higher in PCNL than in others. Interestingly, for > 2 cm stones, PCNL had the highest stone-free rates (SFR) as well as significantly lower auxiliary procedures and retreatment rates than RIRS and ESWL.

Therefore, the authors conclude that PCNL is a safe and effective treatment for large renal stones.

Minor comments

  1. Fig 1: please include in the flow chart the total number of patients analyzed in the qualitative analysis
  2. Table 1: please include stone-free definition as it might be different for different locations
  3. Line 113: the sentence is incomplete
  4. Please detail statistical analysis in materiel and methods section

Major comments

Did the authors analyze any association of the different treatments with age, sex, stone size, and stone location? How the stone-free rate is defined across different location studies? How did the statistics take into account the heterogeneous detection/definitions of stone-free rate? Did the authors evaluate the risk of bias and if yes, with which tools?

Author Response

Dear Reviewer

Thank you for your thoroughly reviewing our manuscript (medicina-1024184) entitled “Effectiveness of Percutaneous Nephrolithotomy, Retrograde Intrarenal Surgery, and Extracorporeal Shock Wave Lithotripsy for Treatment of Renal Stones: A Systematic Review and Meta-Analysis” Also, we are grateful for the chance to revise our manuscript. Our manuscript has been carefully revised according to the reviewers’ comments. Please find our responses to the reviewer’ comments beginning on the next page.

We hope that our revised paper is acceptable for publication in Medicina, and we look forward to receiving your final decision.

Thanks, again.

Sincerely,

*Co-corresponding authors: Joo Yong Lee, MD, PhD, and Seon Heui Lee, PhD

Joo Yong Lee, MD, PhD

Department of Urology, Severance Hospital, Urological Science Institute, Yonsei University College of Medicine, 50-1 Yonsei-ro, Seodaemun-gu, Seoul 03722, Korea

Center of Evidence Based Medicine, Institute of Convergence Science, Yonsei University, Seoul 03722, Korea;

Tel: +82-2-2228-2320; Fax: +82-2-312-2538; E-mail: [email protected]

Seon Hui Lee, PhD

Department of Nursing Science, College of Nursing, Gachon University, 191 Hambangmoe-ro, Yeonsu-gu, Incheon 21936, Korea

Tel: +82-32-820-4235; Fax: +82-32-820-4201; E-mail: [email protected]

# Reviewer 3

This is a meta-analysis based on 37 publications that compared RIRS (retrograde intrarenal surgery), PCNL (percutaneous nephrolithotomy), and ESWL (extracorporeal shock wave lithotripsy) treatments for renal calculus by stone size and analyses treatment efficacy using various indexes like clinical aspects, retreatment rate, and complications.

From their analysis, the authors found that RIRS requires significantly more retreatment procedures than PCLN for > 2 cm stones. The complication was higher in PCNL than in others. Interestingly, for > 2 cm stones, PCNL had the highest stone-free rates (SFR) as well as significantly lower auxiliary procedures and retreatment rates than RIRS and ESWL.

Therefore, the authors conclude that PCNL is a safe and effective treatment for large renal stones.

[Answers]

Thank you for your comment. We also agree with your feedback.

Thank you for your careful consideration.

[Major comments]

Q1. Did the authors analyze any association of the different treatments with age, sex, stone size, and stone location?

A1. Thank you for comment. We did not analyze about age and sex. However, the focus of our study was the results of an additional analysis conducted based on a stone size of 2 cm. Through our research, we have confirmed the success rate and stability of PCNL in renal stones over 2cm. In addition, additional analysis was conducted for lower pole stone. This was reported Table 2 (the summary of results)..

Q2. How the stone-free rate is defined across different location studies?

Q3. How did the statistics take into account the heterogeneous detection/definitions of stone-free rate?

A2, 3. Thank you for comment. The definition between each study of SFRs is described in Table 1. We defined 4mm sized or less as clinically insignificant residual fragments based on the definition of the included studies and previous reference studies. Therefore, regardless of the location of the renal stones, 4mm sized or less was defined as SFR and analyzed.

3.3. Stone-Free Rate

Figure 3 shows a comparison of the SFRs of 37 studies. The definition between each study of SFRs is described in Table 1. We defined 4mm sized or less as clinically insignificant residual fragments based on the definition of the included studies and previous reference studies [57,58]. Therefore, regardless of the location of the renal stones, 4mm sized or less was defined as SFR and analyzed.

Reference

  1. Osman, Y.; Harraz, A.M.; El-Nahas, A.R.; Awad, B.; El-Tabey, N.; Shebel, H.; Shoma, A.M.; Eraky, I.; El-Kenawy, M. Clinically insignificant residual fragments: an acceptable term in the computed tomography era? Urology 2013, 81, 723-726, doi:10.1016/j.urology.2013.01.011.
  2. Seitz, C.; Tanovic, E.; Kikic, Z.; Memarsadeghi, M.; Fajkovic, H. Rapid extracorporeal shock wave lithotripsy for proximal ureteral calculi in colic versus noncolic patients. Eur Urol 2007, 52, 1223-1227, doi:10.1016/j.eururo.2007.02.001.

Q4. Did the authors evaluate the risk of bias and if yes, with which tools?

A4. The contents mentioned are described in chapter 3.2. Thank you.

3.2. Quality assessment and publication bias

The results of the quality assessment based on the Scottish Intercollegiate Guidelines Network checklist are shown in Table 1. Three studies [40,44,51] had a high risk of selection bias, indicating a quality of 1-. Concealment method and blinding were not reported.

The funnel plot included in the meta-analysis is shown in Fig. 2. There was little publication bias in all analyses (Fig. 2).

[Minor comments]

  1. Fig 1: please include in the flow chart the total number of patients analyzed in the qualitative analysis
  2. Table 1: please include stone-free definition as it might be different for different locations
  3. Line 113: the sentence is incomplete
  4. Please detail statistical analysis in materiel and methods section

[Answers]

Thank you for your attentive comments. We have corrected all the points mentioned.

Round 2

Reviewer 1 Report

The authors have addressed all my comments/suggestions. I found their responses quite satisfactory and the new and revised version has been much improved. I now recommend the paper for publication Medicina